# Patients’ Preferences and Willingness to Pay for Solid Forms of Oral Medications—Results of the Discrete Choice Experiment in Polish Outpatients

**DOI:** 10.3390/pharmaceutics12030236

**Published:** 2020-03-06

**Authors:** Marta Kurczewska-Michalak, Przemysław Kardas, Mikołaj Czajkowski

**Affiliations:** 1Department of Family Medicine, Medical University of Lodz, 60, Narutowicza St., 90-136 Lodz, Poland; przemyslaw.kardas@umed.lodz.pl; 2Department of Economics, University of Warsaw, 00-241 Warsaw, Poland; mc@uw.edu.pl

**Keywords:** patient preferences, medication adherence, tablets, capsules, discrete choice experiment

## Abstract

Purpose: Personal preferences play an important role in the patients’ decision process whether to adhere to treatment, or not. The purpose of this study is to assess Polish outpatients’ preferences for solid forms of oral medications across various scenarios. Methods: The convenience sample of 200 outpatients took part in this study. According to the discrete choice experiment (DCE) design, participants were provided with selection of tablets and capsules, in different shapes, sizes, colors, and copayment levels and were asked to state their preferences. DCE results were analyzed using mixed logit (MXL) models. Results: MXL models revealed patients’ willingness to pay for various solid forms of medications. The most preferred combination of drug parameters were: small yellow capsule for chronic treatment, and small yellow long tablet for short-term treatment. Study participants were happy to pay extra 6.52 PLN (≈1.63 EUR) for this drug formulation per month of antihypertensive treatment, and 14.44 PLN (≈3.61 EUR) for this drug formulation per 7-days’ long course of antibiotic treatment, respectively. Conclusion: Results suggest that color, shape, and size of solid form of oral medications are important predictors of patients’ acceptance. It can not only be expressed in monetary value, but also may serve as an important hint for companies designing new drugs, or policymakers who are happy to improve patient adherence with better prescribing, or dispensing. However, our findings are probably country-specific, and further research is necessary to better understand the relationship between solid drug characteristics, and patient’s preferences across countries.

## 1. Introduction

Nonadherence to medication is a widespread phenomenon. According to the World Health Organization, approximately 50% of patients with chronic conditions do not take their drugs as prescribed [1]. This includes various deviations from agreed therapy, i.e., non-initiating, poor execution (implementation), and premature discontinuation of the treatment [2]. There is an evidence showing that this problem has higher than average prevalence in Poland; recent multinational study proved that as many as 58% of Polish patients reported non-adherence to the antihypertensive treatment, while the mean level of non-adherence for nine studied European countries was 44% [3]. Nonadherence leads to a number of negative health and social consequences, and a rapid growth of healthcare costs. Therefore, it is a public health issue of the utmost importance for Polish healthcare system [4].

Numerous factors have been found to affect the level of patient adherence [5]. WHO model groups them into five clusters of patient-, condition-, treatment-, and health system-related factors, as well as social and economic ones [1]. Among these, several treatment-related factors play special role. As soon as the patient has already compared the advantages (necessities) and disadvantages (concerns) of drug taking, and concluded with certain level of willingness to initiate/continue the treatment, these factors stand for either barriers toward, or facilitators of adherence [6]. These particularly include factors important from the patients’ perspective, such as various parameters defining the form of a drug. They are of high practical usefulness in adherence targeting interventions because during dispensation, Polish patients have high level of control over these factors. Currently, wide range of formulations is available for most popular drugs (e.g., web based drug search tool www.doz.pl provides over 20 brands of amlodipine being available on the Polish market) and it is up to the patients to make their own choices when being dispensed drugs from community pharmacies. Originally, this law was set to promote generic substitution, in order to reduce out-of-pocket patient copayment.

In Poland, inpatient drugs are fully covered by the public payer (National Health Fund). Ambulatory treatment patients receive drugs for free in exceptional cases only, for example, if they are seniors aged 75 years and more. The majority of the drugs require co-payment that depends on drug class, patient category, and indication. There are four levels of drug reimbursement: free of charge, lump sum, and co-payment levels of 30% and 50%.

Despite Poland aiming to ensure their citizens’ proper access to safe and effective medicines, the effective out-of-pocket co-payment is still high (more than 60%, on average) [7]. With this economic incentive being in place, along with the obligation for pharmacists to inform patients about the availability of more affordable equivalents [7], patients tend to prefer drugs in generic versions. In 2013, the share of generic drug sales amounted to 57% in terms of volume and 42% in terms of value [8], placing Poland among top users of generic drugs in Europe [8].

Unfortunately, so far, the real impact of drug formulation parameters on patient adherence is understudied, and in particular, no firm data have been collected for the effect of patient’s preferences on the level of medication adherence in Poland. Therefore, the aim of this study is to assess Polish outpatients’ preferences as to the solid forms of oral medications under different treatment scenarios (acute, short-term therapy and chronic, long-term treatment). In order to allow for unbiased in-depth understanding of these preferences, a discrete choice experiment was employed. Having in mind Polish outpatients’ daily experience with co-payment for their drugs, we also decided to calculate patients’ willingness to pay (WTP) for the most preferred versions of drugs, in order to lay foundation for future effective interventions addressing non-adherence.

## 2. Data and Methods

### 2.1. The Discrete Choice Experiment Method

Consumers’ preferences can be modelled based on data from two main sources—revealed or stated choices. The former refers to the case in which individuals’ actual choices are observed, whereas the latter uses choices made in experimentally controlled hypothetical settings [9]. In our case we used patients’ stated preferences, by engaging them in a discrete choice experiment (DCE). This method has been used in a vast number of papers in many fields of applied economics, including environmental, transport, marketing [10], and health studies [11].

In a DCE the good is described as a bundle of individually varied and separately valued attributes (e.g., size, shape, color, form, and price of an oral medicine). The alternatives that are presented to respondents are described in terms of these attributes and their levels. Survey respondents are then asked to choose the best (the most preferred) alternative. These choices can then be analyzed (with the use of the appropriate statistical methods) to formally model mathematical representations of consumer preferences. As a result, it is possible to evaluate the changes in consumer welfare in the case of introducing market changes (e.g., providing a new good) and to predict consumers’ behavior that is related to new goods or alternatives. In addition, identifying the marginal rates of substitution between particular characteristics of a good (including the pecuniary attribute, e.g., the cost for respondents’ household) makes it possible to identify people’s WTP for goods and their characteristics.

In our case, the DCE was used to infer about participants’ preferences as to the solid forms of oral medications, as well as to assess their WTP for particular characteristics of the forms of medications. In order to allow for this, our DCE included a series of sixteen choice questions; each question presenting two alternatives of oral medication forms, defined by five physical attributes (Table 1). 

The attributes included form of oral medication (tablets vs. capsules), shape, size, color, and copayment levels (illustrated by both numbers, and pictures of relevant coins and/or bank notes. Figure 1 presents three examples of choice situations.

Eight choice tasks dealt with chronic treatment (e.g., hypertension), and another eight considered short-term treatment (e.g., antibiotic therapy). The order of respective questions was randomized. The choices were presented to study participants one by one, and the participants were asked to choose the option (medication form at a given price) they prefer, considering its cost. Detailed description of sixteen choice questions set was presented in Appendix A. During the interview the questions considering either chronic or short-term treatment were always rotated in order of two at random. Respondents were always asked to choose one of the option offered. 

### 2.2. Sample

Semi-structured, face-to-face interviews were conducted with 200 Polish primary care outpatients recruited for the survey, and included collection of the basic demographic data, and the DCE exercise. To qualify for inclusion, patients had to be aged 18 years or older, and voluntary agree to participate in the study.

Respondents were distributed almost equally according to gender (51% female) and chronic disease prevalence (51.5% positive), had a median age of 48.8 years. Most of them were employed (63.5%) or pensioners (30.5%). Detailed characteristics are provided in Appendix A. 

### 2.3. Statistical Analysis Framework

Respondents’ choices are modelled using the random utility framework [12]. Respondent *i*’s utility associated with choosing alternative *j* of the *J* available alternatives in choice task *t* can be expressed as:$$U_{ijt}=X_{ijt}b_{i}+c_{ijt}a_{i}+\varepsilon_{ijt},$$
where X represents a vector of alternative-specific attribute levels, *C* is a cost of an alternative, and vector *b* and a are coefficients. The coefficients are indexed by individuals, indicating the possibility of preference heterogeneity but instead of separately estimating the parameters for each individual, we assume that the individual parameters follow specific distributions. This leads to the mixed (random parameters) multinomial logit (MXL) model [13]. 

The MXL model can be rescaled, so that utility function parameters can be directly interpreted as implicit prices (marginal WTPs) for the non-monetary attributes X_ijt_. This “WTP-space” model [14], can be formally described by the following money-metric utility functionThere is a direct translation between asymptotic parameters in models estimated in preference space and WTP space [15] and the two expressions of utility are behaviorally equivalent. Any distribution of parameters in preference space implies some distributions in WTP space, and vice versa. In some cases, however, the resulting distributions can lead to implausible values for WTP or preference parameter estimates [16].
$$U_{ijt}=\alpha_{i}\left(X_{ijt}\frac{b_{i}}{\alpha_{i}}-c_{ijt}\right)+\varepsilon_{ijt}=\alpha_{i}\left(X_{ijt}\beta_{i}-c_{ijt}\right)+\varepsilon_{ijt}$$

We assume that all parameter distributions are normal and the distribution of preference-space (negative of) cost parameter is lognormal.

The model can be estimated using the maximum likelihood method. The probability of an individual choosing an alternative *j* from a set of *C* alternatives (U_ijt_ > U_ikt_, for all k ≠ j) is given by:$$P\left(j|C\right)=\frac{\exp\left(\alpha_{i}\left(X_{ijt}\beta_{i}-c_{ijt}\right)\right)}{\sum_{k=1}^{C}\exp\left(\alpha_{i}\left(X_{ikt}\beta_{i}-c_{ikt}\right)\right)}$$

There is no closed form expression of the above formula for the MXL model, but it can be numerically simulated by averaging over *D* draws from the assumed distributions. We follow Czajkowski and Budziński [17] and use 10,000 scrambled Sobol draws for simulations. The simulated log-likelihood function becomes:$$\log L=\sum_{i=1}^{N}\log\frac{1}{D}\sum_{d=1}^{D}\prod_{t=1}^{T_{i}}\sum_{k=1}^{C}y_{ikt}\frac{\exp\left(\alpha_{i}\left(X_{ijt}\beta_{i}-c_{ijt}\right)\right)}{\sum_{k=1}^{C}\exp\left(\alpha_{i}\left(X_{ikt}\beta_{i}-c_{ikt}\right)\right)},$$
where y_ikt_ is a dummy taking the value 1 if alternative is chosen in choice situation t, and zero otherwise. Maximizing the log-likelihood function in gives estimates for the parameters. The software codes for estimating the MXL model were developed in Matlab and are available at http://github.com/czaj/DCE under Creative Commons BY 4.0 license. The questionnaire, data, software codes, and other Appendix A are available from http://czaj.org/research/supplementary-materials.

## 3. Results

The main results—the estimated means and standard deviations of normally distributed WTP in the sample—are summarized in Table 2.

In the case of short-term treatment (such as antibiotic therapy), respondents were on average willing to pay 2.41 EUR more for medications in the form of long tablets or 1.92 EUR more for capsules, as compared to round tablets. Small-sized medications were strongly preferred, with respondents’ WTP dropping by 2.58 and 4.61 EUR for medium- and large-sized pills, respectively. Respondents’ choices also revealed that they were willing to pay 1.40 EUR more for yellow and 0.33 EUR less for red medications, as compared to the baseline white. 

In the case of chronic treatment the results were similar, however, respondent’ WTP for long tablets and capsules was generally lower (0.46 EUR and 0.96 EUR, respectively). Once again, they strongly preferred small-sized medications. The most preferred color was yellow, followed by white, blue and red being the least preferred. 

Overall, the importance of physical characteristics in both, chronic and short-term treatment scenarios respondents put in the following order: size, form, color. Form was significantly more important in the case of short-term treatment. 

Finally, we note that our respondents’ preferences display substantial unobserved preference heterogeneity, as indicated by relatively large and statistically significant estimates of the standard deviations of the distributions of WTP in the sample. This means that substantial share of patients would be below or above the estimated mean WTP, depending on their individual preferences. To gain some insight into these preferences, Appendix A presents the results of the model in which unobserved preference heterogeneity is dropped and instead the model includes interactions of all attributes with socio-demographic characteristics. Female respondents had significantly stronger negative preferences for larger-sized medications, and in the case of short-term treatment had stronger preferences for long tablets. Interestingly, older respondents were less negative about large-sized pills and of long tablets or capsules rather than round tablets. On the other hand, patients who currently took chronic medications expressed higher WTP for long tablets, were less negative about medium-sized pills and had stronger preferences for yellow-colored pharmaceuticals. These findings were only statistically significant for chronic treatments, however. 

## 4. Discussion

Inconveniences associated with solid form-related attributes, such as difficulties in swallowing because of size and shape, seem to be a well-known problem among patients. Nevertheless, only a few studies have addressed this issue so far. Recent literature review identified 45 studies focusing on patients’ preferences for pharmaceutical preparations. Of these, only 35 investigated dosage form design, and 11 exclusively assessed dosage forms for the oral route. None of these studies used a DCE analysis that provides valid estimates of WTP [18]. 

In particular, there is no such research conducted in Polish population. Therefore, results of this study could be useful for development of national drug policy, especially taking into account the high use of generic medicines in Poland (currently, 57% of market share in terms of volume, [7]).

DCE survey is a systematic method grounded in both economics [15] and psychology [18]. Discrete choice experiment analysis is increasingly used in medical scientific research to quantify patient preferences for attributes of recommended medication [19]. In [20] Mohamed F.A. et al. survey DCE was used to quantify preferences and stated adherence for inhaled antibiotic treatments in cystic fibrosis (CF). Respondents faced five treatment-related choices, evaluating pairs of hypothetical CF treatment profiles. They found that lower frequency of administration, shorter administration times, and milder dry cough appeared to improve stated adherence to antibiotic treatment of CF lung infections. Hauber et al. [21] examined patients’ preferences for attributes of oral antihyperglycemic agents among patients with type 2 diabetes mellitus, and in the other study, the effect of pill burden on dosing preferences, willingness to pay, and likely adherence among the same group of patients [22]. In the first two studies the most important measures of effectiveness were glucose control, medication-related cardiovascular risk, and weight gain, respectively. Weight gain and cardiovascular risk had significant negative effects on likely medication adherence. Next studies pointed at improvements in efficacy as the most valuable for the respondents. Less valued were side effects and dosing, patients willingness to pay for more convenient dosing depended on current dosing burden. 

A DCE study of 2549 patients was carried out by Holmes et al. [23] to assess what influences persistence with medicines. The study found that patients care most about the treatment outcomes, reduced risk of severe (but relatively rare) adverse effects, and dosing frequency. Persistence was therefore related to the willingness to trade potential benefits for reduced harm and increased convenience.

Overgaard and colleagues investigated the swallowability and the patient preferences of tablets and capsules with different sizes, shapes, surfaces, and colors in 331 patients from university teaching hospital. The gelatin capsule was chosen as the most easily swallowed, followed by coated tablet, and the uncoated one [24]. 

In our study results were dependent on the type of the studied therapy (chronic vs. acute treatment). Small capsule was the best choice for chronic therapy, while for short antibiotic treatment patients preferred the small long tablet (however, the difference between long tablet and capsule was not statistically significant). Preferences of medication’s color were, up to a point, for both of the options similar. Our respondents every treatment scenarios chose yellow color as the most preferred and red as the worst for the medications. In Overgaard study [24] the most popular color among the tablets was white, the second was gold and the most disliked was purple. 

In our study, respondents were asked to make choices assuming they required the treatment and hence had to choose of the oral medication options (there was no opt out option). We acknowledge that this may bring consequences for the overall level of estimated WTP values, however, our results are still valid representations of respondents’ relative preferences.

In line with other recent studies, our investigation shows that small size of tablet is associated with the ease of swallowing and it is an important determinant of patients’ choices [25]. Interestingly, older respondents were less negative about large-sized pills and long tablets or capsules rather than round tablets, which may probably result from two different aspects: older patients get used to the particular form of their medication and difficulty of seeing and swallowing small-sized pills by older patients. 

Multiple logistic regression analysis, conducted by Ibrahim et al. indicated that gender, ease of swallowing, and perceptions of the therapeutic benefits of the oral solid drug formulations were significant predictors of capsule preferences [26]. These results seem similar to our chronic treatment results, where the respondents chose capsule as the best solid form of medication. 

In our study, conducted in a convenience sample of Polish outpatients, we proved noticeable heterogeneity of patient preferences as to the oral drugs characteristics and different treatment scenarios (long-term vs. short-term). This observation led us to the idea that further research is necessary to fill the gap in understanding of basic determinants of patients behavior regarding drug taking.

There are several limitations of this study that should be considered when interpreting the results. In DCEs, patients evaluate hypothetical treatments and thus, differences may arise between their stated and actual choices. [27] discusses the possible treatments of this uncertainty. Moreover, while we used 12 choice tasks per respondent, we acknowledge potential anchoring effects associated with the first treatments considered and the first choice tasks seen. Even though the effects of ordering effects in DCEs have been found to be negligible [28], one could always limit the analysis to the first choice of each respondent. Next, we have studied patients’ preferences only, and not the final effect of these preferences in terms of patient adherence to medication, and health outcomes. This, however, is not a major disadvantage, as the link between patients’ preferences and treatment execution was observed for many times. Second, this study sample was not representative and number of participants, who took part in the research was only 200. However, this study was the first study of patients’ preferences based on DCE method. Therefore, an appropriate sample size could not be calculated a priori. Finally, patients’ preferences as to the drug forms seem to vary across geographical locations [17]. Therefore, the applicability of our results might be limited outside Poland. 

With all these limitations in hand, we are convinced that the results of this study possess high applicability. Knowledge of patients’ preferences provide an evidence to help designing interventions effective in improving patient adherence. For example, they may be used by the pharmaceutical industry to make the drugs more patient-friendly. They may also help better prescribing, acknowledging patients’ preferences during the process of selection of medication for individual use. Finally, they could also enable pharmacy-based interventions. Knowing patients’ preferences, and using generic substitution (which is available according to the law in a number of countries, including Poland), pharmacists could be able to offer their patients drugs that are better tailored toward their needs and preferences.

## 5. Conclusions

The results of our research show that physical characteristics of oral medications such as color, shape, and size are significant determinants of patients’ acceptance. Not only they can be expressed in monetary values, but may also serve as an important suggestion for pharmaceutical companies designing new drugs in order to assure best possible adherence. In Poland, due to the generic substitution there is a wide range of drug formulations available for the patients. Therefore patients are free to take their own choices when being dispensed drugs from community pharmacies. Thus, their preferences may serve as the basis of pharmacy-based interventions within the so-called pharmaceutical care, aiming to improve adherence at the individual level. However, our findings are probably country/culture-specific, and further research is necessary to better understand the relationship between solid drug characteristics and patient preferences across the countries.

## Figures and Tables

**Figure 1 pharmaceutics-12-00236-f001:**
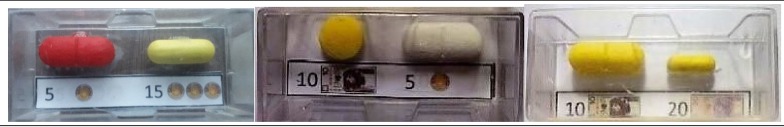
Three examples of the choice tasks presented to study participants.

**Table 1 pharmaceutics-12-00236-t001:** Description of medications presented to study participants.

**Size**	Small—5 mm
	Medium—10 mm
	Large—15 mm
**Shape**	Round ^$^
	Long
**Color**	White
	Yellow
	Blue
	Red
**Form**	Uncoated tablet
	Capsule
**Copayment level (PLN *)**	5 PLN
	10 PLN
	15 PLN
	20 PLN

^$^ not applicable to capsules, * PLN—Polish zloty, 1 PLN ≈ 0.25 EUR.

**Table 2 pharmaceutics-12-00236-t002:** The results of the mixed logit (MXL) model, representing patients willingness to pay (WTP) (in EUR per week) for various forms of short-term and chronic treatment medications.

Attributes	Short-Term Treatment		Chronic Treatment	
	Mean (st. err.)	Std. Dev. (st. err.)	Mean (st. err.)	Std. Dev. (st. err.)
Form—long tablet (vs. round tablet)	2.41 *** (0.15)	0.69 *** (0.10)	0.46 *** (0.05)	0.03 (0.04)
Form—capsule (vs. round tablet)	1.92 *** (0.17)	0.59 *** (0.09)	0.96 *** (0.06)	0.35 *** (0.07)
Size—medium (vs. small)	−2.58 *** (0.14)	0.99 *** (0.15)	−2.63 *** (0.06)	0.73 *** (0.09)
Size—large (vs. small)	−4.61 *** (0.09)	2.11 *** (0.11)	−5.29 *** (0.13)	2.34 *** (0.30)
Color—yellow (vs. white)	1.40 *** (0.15)	0.02 (0.11)	0.67 *** (0.10)	0.13 (0.09)
Color—blue (vs. white)	0.06 (0.09)	0.06 (0.07)	−0.65 *** (0.06)	0.01 (0.06)
Color—red (vs. white)	−0.33 ** (0.14)	0.55 *** (0.08)	−1.11 *** (0.07)	0.38 *** (0.06)
Cost (EUR)	0.30 (0.24)	1.91 *** (0.37)		

Notes: ***, **, and * indicate 1%, 5%, and 10% significance levels, respectively.

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
