# Peer review of "Patients’ Preferences and Willingness to Pay for Solid Forms of Oral Medications—Results of the Discrete Choice Experiment in Polish Outpatients"

_pharmaceutics, 2020, doi:10.3390/pharmaceutics12030236_

Round 1

Reviewer 1 Report

In this manuscript the authors have reported the patients’ preferences of different drug treatments. Methodology seems correct and although more participants would have improved the results, the conclusions are interesting. Some minor typographical mistakes have to be corrected while editing.

Author Response

In this manuscript the authors have reported the patients’ preferences of different drug treatments. Methodology seems correct and although more participants would have improved the results, the conclusions are interesting. Some minor typographical mistakes have to be corrected while editing.

Thank you for the positive evaluation of our work. In the revised version of the manuscript we tried to correct any remaining mistake

Reviewer 2 Report

The study conducted by Marta Kurczewska-Michalak, Przemysław Kardas, Mikolaj Czajkowski regarding " Patients’ preferences and willingness to pay for solid forms of oral medications – results of the discrete choice experiment in Polish outpatients." is of interest. It describes that personal preferences play an important role in the patients’ decision process.

Some observations:

Data and Methods: I think authors should describe sixteen choice questions.

Discussion: “Interestingly, older respondents were less negative about large-sized pills and of long tablets or capsules rather than round tablets.” Older adults have trouble swallowing medicine. The result doesn't make sense, so the author may need to explain why. One more question, why are people not concerned about drug costs?

The use of MeSH descriptors is recommended as keywords.

Author Response

  1. Data and Methods: I think authors should describe sixteen choice questions.

  • In lines 109 – 111 we added an explanation and the detailed description of the choice questions was added in supplementary material.

  1. Discussion: “Interestingly, older respondents were less negative about large-sized pills and of long tablets or capsules rather than round tablets.” Older adults have trouble swallowing medicine. The result doesn't make sense, so the author may need to explain why.

  • In lines 225 – 229 we explained probable reasons of such results.

  1. One more question, why are people not concerned about drug costs?

  • Indeed, we find that respondents are concerned about the cost of the drugs – the cost attribute was a significant determinant of respondents’ choices. We do not present it in our estimates, as in WTP-space models the cost parameter is normalized to 1 (it is used as a reference/denominator, to express trade-offs for other attributes). Indeed, we find that these trade-offs are (1) finite and (2) statistically significantly different than zero.

  1. The use of MeSH descriptors is recommended as keywords.

  • Keywords were changed according to MeSH.

Reviewer 3 Report

This paper presents the results from the stated preference discrete choice experiments applied in health area, specifically, to derive the estimates of marginal willingness to pay values for five attributes of pills, including the cost. Preferences are elicited from the Polish outpatients for a medication orally used for short-term, or chronic treatment. The authors use adequate method (DCE) and proper econometric model (MXL in the preference-space) to estimate the WTP values. They present their results clearly. This manuscript presents a nice piece of applied research and I believe it can contribute to relevant literature. I therefore can recommend this manuscript for publication in Pharmaceutics after minor revisions I describe below.

I have only a few comments related to the design and estimation strategy.

1) The authors use in total sixteen choice tasks (that is quite many), when eight choice tasks dealt with chronic treatment while another eight dealt with short-term treatment. It is not clear whether the choice task dealing with chronic treatment were always asked at first, i.e. before the questions dealing with short-term treatment, or whether they rotated the order of the two at random (what they should do). It may be true that quite small sample size did not allow them to do so. This should be however clarified when the design is described. Moreover, a potential bias of the WTP estimates for the short-term treatment coming from the first exercise shall be discussed in Discussion part (since the only clean ones would be the WTP estimates for the chronic treatment, in the case of no-rotated order).

2) Potentially, responses on the first eight questions may affect the next eight responses. If they rotated the order the random, I recommend to run the models separately for the short-term and chronic treatment, when only responses from the first eight questions are used (i.e. n=100).

3) I do not know whether and how the two choice cards differ (used for the short-term and chronic treatment), but I suppose the two were very similar, maybe identical. If it is the case, did respondents consider the first eight and next eight are actually valuing two different goods? I recommend to run the MXL model merging the data from the two goods and investigate whether the preference parameter for the pill attributes differs for the two goods. Moreover, this strategy may reveal whether the two preference parameters for the same attribute are the same or different.

4) I guess there was not a status quo (opt out) option offered in the experiment. If it is the case, then the authors used the forced question, and they might discuss consequences of this strategy in Discussion part.

Minor comment:
One of the examples of choice situations presented in Figure 1 shows the cost is 40, while the cost levels, as reported in Table 1, are 5, 10, 15, 20 PLN. I guess there used this card either in the presurvey or in the pilot. Anyways, they might replace the example by actual cards or revise the table.

Author Response

  1. The authors use in total sixteen choice tasks (that is quite many), when eight choice tasks dealt with chronic treatment while another eight dealt with short-term treatment. It is not clear whether the choice task dealing with chronic treatment were always asked at first, i.e. before the questions dealing with short-term treatment, or whether they rotated the order of the two at random (what they should do). It may be true that quite small sample size did not allow them to do so. This should be however clarified when the design is described.

  • In lines 109 – 111 we added a comment that now makes it clear that we rotated the order of considered treatments.

  1. Moreover, a potential bias of the WTP estimates for the short-term treatment coming from the first exercise shall be discussed in Discussion part (since the only clean ones would be the WTP estimates for the chronic treatment, in the case of no-rotated order).

  • We did rotate the treatment order. We now acknowledge potential anchoring effects associated with the first choice tasks and the first treatments considered (lines 220 – 223).

  1. Potentially, responses on the first eight questions may affect the next eight responses. If they rotated the order the random, I recommend to run the models separately for the short-term and chronic treatment, when only responses from the first eight questions are used (i.e. n=100).

  • In lines 109 – 111 we added a detailed description of study design and questions rotation. We did not observe statistically significant effects associated with the order of treatments (however, we acknowledge that this could be due to the limited sample size).

  1. I do not know whether and how the two choice cards differ (used for the short-term and chronic treatment), but I suppose the two were very similar, maybe identical. If it is the case, did respondents consider the first eight and next eight are actually valuing two different goods? I recommend to run the MXL model merging the data from the two goods and investigate whether the preference parameter for the pill attributes differs for the two goods. Moreover, this strategy may reveal whether the two preference parameters for the same attribute are the same or different.

  • The experimental design used for the treatments differed (we used efficient design with priors based on preferences for each treatment observed in the pilot study), however, the choice tasks used the same attributes and attribute levels. We do find statistically significant differences between treatment-specific preference parameters – note the differences between parameter estimates reported in Table 2.

  1. I guess there was not a status quo (opt out) option offered in the experiment. If it is the case, then the authors used the forced question, and they might discuss consequences of this strategy in Discussion part.

  • In lines 220 – 223 we discussed probable consequences of our strategy. Since in our case the treatment necessarily required administering oral medication, the forced choice was a natural choice.

  1. Minor comment:

One of the examples of choice situations presented in Figure 1 shows the cost is 40, while the cost levels, as reported in Table 1, are 5, 10, 15, 20 PLN. I guess there used this card either in the presurvey or in the pilot. Anyways, they might replace the example by actual cards or revise the table.

  • A more detailed description of choice situations is now presented. The earlier example was indeed taken from the pilot study – it is now replaced by actual choice tasks used in the main study.

Reviewer 4 Report

Overall, this is more an empirical study. The topic is interesting and important since the nonadherence to medication for outpatients in Poland is much higher than that in other European countries. The article is well structured and written. The DCE method applied in this study is appropriate for this study and is also well conducted. Results are discussed well and implications could provide potential suggestions for policy makers.

Here are some comments in detail:

  • For Data and Methods:

Table 1 shows the attributes and the associated levels of drugs. But why those drugs should have these attributes and levels? It worth to have both theoretical and empirical explanations.

The choice sets are not presented in the article, so I am not sure “whether not to choose any” has been added to the choice.

Equations in this part is presented not well which some part of the equations are not readable. Equations need to be formatted carefully please.

  • For Results:

Theoretical mixed logit model includes individual heterogeneity which assumes that each individual have different preferences for each different attributes. However, the results for this part is neither presented not discussed, which suppose to be part of your topic.

  • For Discussion:

In side this part, uncertainty about the stated preferences has been mentioned as a disadvantage. The following article might provide you some thoughts on this one:

Jia Wang, Jiaoju Ge* (2016) “Alternative approaches to treat respondent uncertainty in contingent willingness to pay estimation: a theoretical and empirical analysis.” China Agricultural Economic Review. 8(3) 412-429

Author Response

For Data and Methods:

  1. Table 1 shows the attributes and the associated levels of drugs. But why those drugs should have these attributes and levels? It worth to have both theoretical and empirical explanations.

  • The attribute levels selected for our study correspond to the usual form of oral medications administered to patients in Poland. Similarly, the cost levels reflect the possible range of costs outpatients face when receiving the treatment.

  1. The choice sets are not presented in the article, so I am not sure “whether not to choose any” has been added to the choice.

  • The opt-out was not available to respondents – Figure 1 presents three example choices used in the study. The choices were made assuming a patient required the treatment (and hence he or she needed to choose one of the available forms of medications). Some comments were added in lines 109 – 111 and 220 – 223 to make this clearer.

  1. Equations in this part is presented not well which some part of the equations are not readable. Equations need to be formatted carefully please.

  • Equations were corrected (the earlier problems were likely a result of font incompatibility).

For Results:

  1. Theoretical mixed logit model includes individual heterogeneity which assumes that each individual have different preferences for each different attributes. However, the results for this part is neither presented not discussed, which suppose to be part of your topic.

  • Indeed, we find substantial unobserved preference heterogeneity associated with our choice attributes, as evidence by statistically significant and relatively large estimates of standard deviations. We now also discuss this in the Results section (lines 165 – 171).

For Discussion:

  1. Inside this part, uncertainty about the stated preferences has been mentioned as a disadvantage. The following article might provide you some thoughts on this one:

Jia Wang, Jiaoju Ge* (2016) “Alternative approaches to treat respondent uncertainty in contingent willingness to pay estimation: a theoretical and empirical analysis.” China Agricultural Economic Review. 8(3) 412-429

Thank you for this valuable suggestion – we now use this reference in our discussion of uncertain